# Hyphenated High Performance Liquid Chromatography–Tandem Mass Spectrometry Techniques for the Determination of Perfluorinated Alkylated Substances in Lombardia Region in Italy, Profile Levels and Assessment: One Year of Monitoring Activities During 2018

**Salvatore Barreca** [1],*[ID]**, Maddalena Busetto** [1]**, Luisa Colzani** [1]**, Laura Clerici** [1]**, Valeria Marchesi** [2]**, Laura Tremolada** [2]**, Daniela Daverio** [1] **and Pierluisa Dellavedova** [1]

[1]  Agenzia Regionale per la Protezione dell'Ambiente della Lombardia (ARPA Lombardia); Settore Laboratori: U.O. Laboratorio di Milano, Sede laboratoristica di Monza e Brianza via Solferino n° 16, 20900 Monza (MB), Italy; m.busetto@arpalombardia.it (M.B.); l.colzani@arpalombardia.it (L.C.); l.clerici@arpalombardia.it (L.C.); d.daverio@arpalombardia.it (D.D.); p.dellavedova@arpalombardia.it (P.D.)

[2]  Agenzia Regionale per la Protezione dell'Ambiente della Lombardia (ARPA Lombardia); Settore Monitoraggi Ambientali UO Risorse Idriche: Programmazione e Coordinamento Via I. Rosellini, 17 20124 Milano, Italy; v.marchesi@arpalombardia.it (V.M.); l.tremolada@arpalombardia.it (L.T.)

*  Correspondence: s.barreca@arpalombardia.it

**Abstract:** In this research paper, we report a hyphenated technique based on ultra-high performance liquid chromatography–tandem mass spectrometry for the determination of twelve Perfluorinated Alkylated Substances in surface and groundwater samples from Lombardia Region during the monitoring activities in 2018 as new emerging and toxic pollutants. A green analytic method, developed by using an online Solid Phase Extraction coupled with UHPLC-MS/MS and previously validated, was applied for 4992 determinations conducted on 416 samples from 109 different sampling stations. Among the results, PFOS, PFOA, PFBA, PFBS, PFPeA and PFHxA were identified as the most abundant analytes detected. PFASs concentrations, in most cases, were below the limits of quantification and, in the cases where the limits of quantification have been exceeded, the values found were lower than Italy directive. PFOS is an exception and in fact this compound was detected in 76% of analyzed samples (surface and ground waters). Solid phase extraction with high performance liquid chromatography–tandem Mass Spectrometry has proved to be a very good Hyphenated techniques able to detect low concentrations of pollutants in surface and groundwater samples.

**Keywords:** PFAS determination; water monitoring activities; UHPLC-MS/MS; hyphenated techniques

## 1. Introduction

In the last twenty years, a wide variety of pollutants have been found in soils, sediments and water [1–3]. For this reason, the environmental community has made efforts to improve monitoring activities, especially in the context of water [4,5] and emerging pollutants. In this regard, the hyphenated techniques offer sensitivity and robustness in order to achieve quality performances as required from regulations.

Today, an important class of pollutants is represented by Perfluorinated Alkylated Substances. The term PFASs (Perfluorinated Alkylated Substances) refers to a family of synthetic organic compounds,

in which all the hydrogens of the hydrocarbon backbones are substituted with fluorine atoms [6]. It is a category of so-called "emerging" compounds, whose presence in the environment has only relatively recently been highlighted and whose determination has become technically possible in the different matrices.

Their key characteristic is that they have one of the strongest chemical bonds (C–F) in their carbon chain, which makes them very stable, and they cannot degrade naturally or when exposed to heat, acids, or oxidation.

Such compounds are widely used in everyday life, for example in adhesives, polishes, paints, waxes, textiles, in stain/water/grease repellents in carpets and clothing or in cooking utensils such as non-stick coatings [7–9]. In addition, former research suggests that the impact of exposure from consumer products is usually lower than the impact of exposure to contaminated drinking water or food, such as fish [10,11]. Generally, PFASs accumulate in the human body and their levels decrease slowly over time.

Some studies in humans have shown that certain PFASs may have an impact on developing fetus, may decrease fertility, interfere with the natural hormones of the body, increase cholesterol, affect the immune system, and increase cancer risk [12,13].

Due to their widespread diffusion, persistence, ability to bioaccumulate, and toxicological properties, PFASs have attracted increasing attention from the international scientific community and competent European regulatory authorities, especially for PFOS and PFOA (Perfluorooctane sulfonate and perfluorooctanoic acid respectively), which were added at the list of "dangerous and priority" substances be monitored in water bodies (Directive 2013/39 / EU).

In recent years, Environmental Protection Agency of Lombardia Region (ARPA Lombardia) form Italy has invested economical sources in monitoring activities, and recently, laboratories have developed an environmentally green analytical chemistry methods, by hyphenated techniques. In detail, the method consists of a concentration by online solid phase extraction coupled with ultra-high-performance liquid chromatography–tandem mass spectrometry, in order to quantify PFASs at very low limit of quantifications [14].

During 2017 ARPA Lombardia started monitoring PFAS in some areas of the regional territory (Serio's basin, Olona's basin and Mantua-Brescia area), and in 2018 began a systematic monitoring in waterways and groundwater.

The aim of this study was to assess PFASs concentrations in Lombardia Region in Italy in order to estimate the pollution levels in several water samples.

## 2. Experimental

### 2.1. Chemicals and Materials

Native and $^{13}$C mass-labeled standard methanolic solutions were purchased from Wellington Laboratories. In detail, the standard mix solution PFAC-MXB was purchased with chemical purities of >98% and a concentration of 2000 ng mL$^{-1}$ in Methanol/Water <1% for every individual perfluoroalkylcarboxylic acid and perfluoroalkylsulfonate. All compounds were purchased in methanol mix solution and in linear form (no isomeric compounds were reported).

Stock solutions were prepared in methanol and stored at −18 °C in amber glassware. Methanol HPLC grade was purchased from Sigma Aldrich. NH$_4$OH concentrate at 28% v/v solution was purchased from Sigma Aldrich.

High purity water was prepared using a Millipore Milli-Q purification system. Mass-labeled solution and calibration solutions were prepared by serial dilution of stock solutions in tap water (S. Anna). For the SPE on-line pre-concentration and clean-up experiments Oasis On-Line SPE Symbiosis Prospeky-2$^{TM}$ Cartridge WAX 10 × 2 mm made in Netherlands was used, relying upon the good recoveries reported in the existing literature by using Oasis WAX-SPE cartridges including short-chain (C4–C6) compounds.

Two LC Column Poroshell 120 EC-C18 3 × 50 mm, 2.7 μm made in the USA were used for eluent clean up.

A Gemini C18 3 μm 100 × 3 mm made in Netherlands by Phenomenex operating in a pH value ranging from 1 to 12 was used as a chromatographic separation column. Samples were collected in a 50 mL propylene tube and refrigerated at 4 °C before the analyses. Analyses were performed within 15 days.

*2.2. Solvent Cleanup*

Methanol and Milli-Q water were pre-cleaned by connecting separate on-line scrubber columns for each mobile phase. In this context, two columns were used. The first one was positioned before the injection valve to clean up the eluent used for SPE online, and the second one was used to clean up the eluent employed in chromatographic separation.

*2.3. Liquid Chromatograph and Mass Spectrometric Instruments*

Analysis was performed using Ultra High-Performance Liquid Chromatograph (UHPLC) consisting of binary and quaternary pump Shimadzu/Nexera LC pump, a DGU-20A 5R degassing unit, a SIL-30AC autosampler equipped with a 5000 L loop, a CTO/20AC thermostated column compartment and a CBM-20A made in the USA module controller were used as other components.

The Nexera LC system was coupled to a 6500 Q-Trap mass spectrometer (Sciex) made in the USA, equipped with a Turbo V® interface by an ESI probe.

Elution conditions and Mass Spectrometry optimal parameters were reported in a previous paper concerning method validation [14].

Several MS parameters used are reported in Tables 1 and 2.

**Table 1.** Parameter used in ESI-MS/MS.

| Source | Unit | Value |
|---|---|---|
| Curtain Gas (CUR) | psi | 25 |
| Collision Gas | - | High |
| Ion Spray Voltage (IS) | V | −4500 |
| Temperature TEM | °C | 450 |
| Ion Source Gas (GS1) | psi | 45 |
| Ion Source Gas (GS2) | psi | 55 |
| Entrance Potential (EP) | V | −10 |
| Collision Exit potential (CXP) | V | −15 |

**Table 2.** m/z, declustering potential and collision energy used.

| Analyte | Q1 Precursion Ion [M−H]⁻ (m/z) | Q3 Product Ion (m/z) | Declustering potential (DP) V | Q2 Collision Energy (CE) V |
|---|---|---|---|---|
| PFBA-1 | 213 | 169 | −10 | −14 |
| PFPeA-1 | 263 | 219 | −10 | −12 |
| PFBS-1 | 299 | 80 | −40 | −70 |
| PFBS-2 | 299 | 98 | −40 | −40 |
| PFHxA-1 | 313 | 269 | −10 | −13 |
| PFHxA-2 | 313 | 119 | −10 | −30 |
| PFHpA-1 | 363 | 319 | −10 | −15 |
| PFHpA-2 | 363 | 169 | −10 | −20 |
| PFHxS-1 | 399 | 79.9 | −40 | −100 |
| PFHxS-2 | 399 | 98.9 | −40 | −76 |
| PFOA-1 | 413 | 369 | −10 | −15 |
| PFOA-2 | 413 | 169 | −10 | −20 |

**Table 2.** *Cont.*

| Analyte | Q1 Precursion Ion [M−H]⁻ (m/z) | Q3 Product Ion (m/z) | Declustering potential (DP) V | Q2 Collision Energy (CE) V |
|---|---|---|---|---|
| PFNA-1 | 463 | 419 | −10 | −15 |
| PFNA-2 | 463 | 168 | −10 | −25 |
| PFOS-1 | 499 | 80 | −60 | −120 |
| PFOS-2 | 499 | 99 | −60 | −100 |
| PFDA-1 | 513 | 469 | −10 | −15 |
| PFDA-2 | 513 | 218 | −10 | −22 |
| PFUnDA-1 | 563 | 519 | −10 | −15 |
| PFUnDA-2 | 563 | 269 | −60 | −140 |
| PFDoDA-1 | 613 | 569 | −10 | −15 |
| PFDoDA-2 | 613 | 269 | −10 | −20 |
| MPFBA * | 217 | 171 | −10 | −14 |
| $M_5$PFPeA * | 268 | 233 | −10 | −12 |
| $M_3$PFBS * | 302 | 99 | −40 | −70 |
| $M_5$PFHxA * | 318 | 273 | −10 | −13 |
| $M_4$PFHpA * | 367 | 322 | −10 | −20 |
| $M_3$PFHxS * | 402 | 99 | −40 | −100 |
| $M_8$PFOA * | 421 | 376 | −10 | −14 |
| $M_8$PFOS * | 507 | 99 | −60 | −100 |
| $M_9$PFNA * | 472 | 427 | −10 | −15 |
| $M_6$PFDA | 519 | 474 | −10 | −15 |
| $M_7$PFUdA | 570 | 525 | −10 | −15 |
| MPFDoDA | 615 | 570 | −10 | −15 |

### 2.4. Sample Preparation

Samples were collected in a 50 mL Polypropylene tube (purchased from CPS Analitica for Chemistry, Milan, Italy).

Ten mL of samples were transferred to a 10 mL vial, then 0.1 mL Internal Standard (IS) solutions at 5000 ng L⁻¹ were added in order to obtain a final IS concentration of 50 ng L⁻¹; they were manually mixed and stored in a refrigerated autosampler system at 4 °C. 5 mL of sample was injected by using a SPE online system as discussed in previous paper [14]. The samples were injected directly without filtration step.

### 2.5. Method Validation and Quality Assurance

The analyses were performed by using a method developed and previously reported in the scientific literature [14].

Briefly, method performances were evaluated by estimating quantification limits (LOQs) and linearity.

Method was validated in term of quantification limits, accuracy and precision as directive required [15]. Analyses were conducted by using real surface water samples.

The quantification limits (LOQs) were estimated by 10-time standard deviation calculated at first level of calibration curve obtained in surface water sample.

Table 3 summarizes the results obtained for these performance parameters. Accuracy and repeatability were calculated on data set of six analyses conducted on water spiked samples.

Linearity ranges were obtained from 0.2–250 to 5–250 ng L⁻¹ depending on the different analyte considered.

PFBA shows a linear range from 5 to 250 ng L⁻¹. Due to its short alkyl chain, chemical physical properties of PFBA are very different compared to other PFASs compounds. In this regard, PFBA is adsorbed less than other PFASs in SPE online and in this context the first level corresponding to 5 ng L⁻¹. In any cases, by comparing this method with other direct injection or SPE methods, a good

detection limit for PFBA was achieved. For the most PFASs, good correlation can be obtained in calibration curve range from 1 to 250 ng L$^{-1}$.

For PFOS, one of the most important analytes monitored, linearity of method was evaluated at low levels over concentration range from 0.2 to 250 ng L$^{-1}$ finding a linear response (see correlation coefficients in Table 2). Generally, in literature detection limit for PFOS, by direct injection or SPE, ranged from 5 to 10 ng L$^{-1}$ and only few cases reported LOD at level of 1 ng L$^{-1}$. In this context, quantification limits of 0.2 ng L$^{-1}$ without sample preparations, can be considered an interesting analytical result.

Blank and quality control at 10 ng L$^{-1}$ were analyzed every ten samples in order to ensure the best instrument performance. The quality control samples were prepared by mixing the standard solution with Milli-Q water at final concentration of 50 ng L$^{-1}$ and spiked with 50 ng L$^{-1}$ of the internal standard mix. Analysis was performed after the linear calibration curve (linearity ranges were obtained from 0.2–250 ng L$^{-1}$ for PFOS, from 10 to 250 ng L$^{-1}$ for PFUnA and PFDoA, from 5 to 5–250 ng L$^{-1}$ for the other analytes. The quantitation was performed using the MultiQuantTM 3.0.3 software (SCIEX, Framingham, MA, USA).

**Table 3.** Limit of quantification (LOQs), linearity range, R$^2$, Accuracy, and Repeatability parameters evaluated at LOQ.

| Compound | LOQ (ng L$^{-1}$) | Linear range (ng L$^{-1}$) | R$^2$ | Accuracy % | Repeatability CV% |
|---|---|---|---|---|---|
| PFBA | 5 | 5–200 | 0.999 | 101.1 | 10.7 |
| PFPeA | 1 | 1–250 | 0.999 | 91.7 | 10.6 |
| PFBS | 1 | 1–250 | 0.999 | 95.6 | 11.5 |
| PFHxA | 1 | 1–250 | 0.999 | 97.4 | 8.28 |
| PFHpA | 1 | 1–250 | 0.999 | 97.2 | 10.4 |
| PFHxS | 1 | 1–250 | 0.999 | 88.9 | 9.7 |
| PFOA | 1 | 1–250 | 0.999 | 99.9 | 11.5 |
| PFNA | 1 | 1–250 | 0.999 | 93.0 | 13.8 |
| PFOS | 0.2 | 0.2–250 | 0.999 | 85.0 | 12.4 |
| PFDA | 5 | 5–250 | 0.999 | 81.0 | 12.1 |
| PFUnDA | 5 | 5–250 | 0.999 | 107.2 | 11.3 |
| PFDoA | 5 | 5–250 | 0.999 | 107.4 | 9.5 |

## 3. Results and Discussion

A total of 4992 determinations of 416 samples (Tables 4 and 5) taken from 109 sampling stations (57 from ground water and 52 for surface water) located in Lombardia region were investigated (see Figure 1, Tables 4 and 5). The monitoring frequency was choice depending on point of pressures. 68% of the samples were surface water and 32% were ground water. PFASs were detected in 24% of determinations, PFOA and PFOS were detected most frequently.

PFASs concentrations were, in most cases, below the limits of quantification and, in the cases where the limits of quantification have been exceeded, the values found were lower than the Italian Directive No. 172/2015 (from 100 to 3000 ng L$^{-1}$). PFOS is an exception and in fact this compound was detected in 76% of analyzed samples (surface and groundwater) (see Table 6). This data highlights the directly or indirectly (by using consumer products) PFOS use in Lombardia Region.

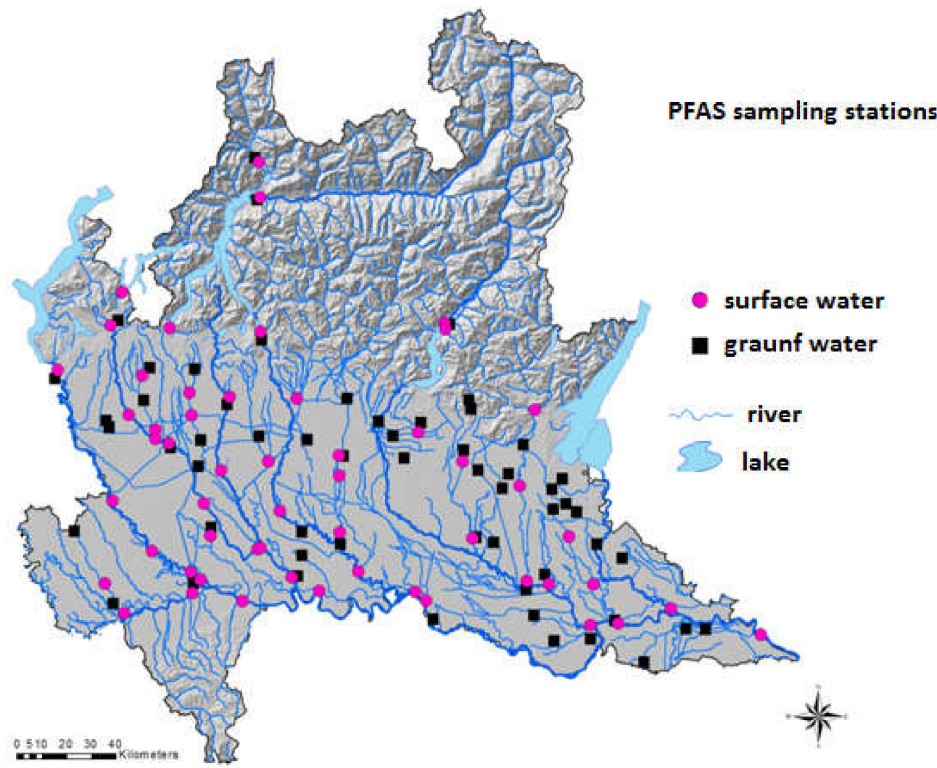

**Figure 1.** Sampling stations located in Lombardia Region in Italy.

**Table 4.** Surface water sampling stations investigate.

| River basin | Surface water name | City | Monitoring frequency |
| --- | --- | --- | --- |
| Brembo | Dordo | Filago | 4 |
| Oglio sopralacuale | Oglio | Costa Volpino | 6 |
| Serio | Serio | Mozzanica | 6 |
| Chiese sublacuale | Chiese | Villanuova sul Clisi | 6 |
| Chiese sublacuale | Chiese | Montichiari | 6 |
| Mella | Fiume | Flero | 4 |
| Lago d'Iseo (Sebino) | Italsider | Pisogne | 6 |
| Mella | Mella | Pralboino | 6 |
| Oglio sublacuale | Seriola Nuova di Chiari | Rovato | 4 |
| Olona-Lambro Meridionale | Antiga | Limido Comasco | 4 |
| Seveso | Seveso | Fino Mornasco | 3 |
| Seveso | Seveso | Vertemate | 3 |
| Adda prelacuale | Adda | Gera Lario | 6 |
| Lago di Como (Lario) | Cosia | Como | 6 |
| Adda sublacuale | Adda | Pizzighettone | 6 |
| Po | Morbasco | Gerre de' Caprioli | 4 |
| Po | Po | Cremona | 6 |
| Serio | Serio | Sergnano | 4 |
| Serio | Serio | Montodine | 6 |
| Adda sublacuale | Adda | Calolziocorte/Olginate | 6 |
| Adda sublacuale | Adda | Montanaso Lombardo | 4 |
| Lambro | Lambro | S. Angelo Lodigiano | 4 |
| Lambro | Lambro | Orio Litta | 6 |
| Olona-Lambro Meridionale | Lambro M. | S. Angelo Lodigiano | 4 |
| Po | Po | Somaglia | 6 |
| Lambro | Lambro | Lesmo | 6 |
| Seveso | Seveso | Lentate | 3 |
| Seveso | Terrò | Seveso | 4 |
| Olona-Lambro Meridionale | Bozzente | Lainate | 4 |

**Table 4.** *Cont.*

| River basin | Surface water name | City | Monitoring frequency |
|---|---|---|---|
| Adda sublacuale | La Molgora | Truccazzano | 4 |
| Lambro | Lambro | Peschiera Borromeo | 4 |
| Olona-Lambro Meridionale | Lambro M. | Locate Triulzi | 4 |
| Olona-Lambro Meridionale | Olona | Legnano | 4 |
| Olona-Lambro Meridionale | Olona | Rho | 4 |
| Olona-Lambro Meridionale | Olona | Pero | 6 |
| Seveso | Seveso | Paderno Dugnano | 4 |
| Ticino sublacuale | Ticino | Abbiategrasso | 4 |
| Chiese sublacuale | Chiese | Canneto sull'Oglio | 6 |
| Mincio | Mincio | Roncoferraro | 4 |
| Oglio sublacuale | Oglio | Marcaria | 6 |
| Mincio | Osone Vecchio | Castellucchio | 4 |
| Po | Po | Borgo Virgilio | 6 |
| Po | Po | Sermide | 6 |
| Mincio | Seriola M. - Osone Nuovo | Ceresara | 4 |
| Oglio sublacuale | Tartaro Fuga | Acquanegra sul Chiese | 4 |
| Agogna | Agogna | Mezzana Bigli | 6 |
| Agogna | Erbognone | Ottobiano | 6 |
| Olona-Lambro Meridionale | Olona | Lardirago | 4 |
| Po | Po | Rea | 4 |
| Po | Po | Arena/Spessa Po | 6 |
| Ticino sublacuale | Ticino | Bereguardo | 4 |
| Ticino sublacuale | Ticino | Pavia | 4 |
| Ticino sublacuale | Ticino | Travacò Siccomario | 6 |
| Mera | Mera | Samolaco | 6 |
| Lago di Lugano (Ceresio) | Bolletta | Porto Ceresio | 6 |
| Olona-Lambro Meridionale | Olona | Varese | 4 |
| Ticino sublacuale | Ticino | Golasecca | 6 |

**Table 5.** Ground water sampling stations investigate.

| City | Monitoring frequency |
|---|---|
| Grassobbio | 3 |
| Isso | 1 |
| Treviglio | 3 |
| Bedizzole | 1 |
| Brescia | 3 |
| Calvisano | 3 |
| Chiari | 3 |
| Comezzano - Cizzago | 3 |
| Concesio | 1 |
| Gambara | 3 |
| Lonato Del Garda | 1 |
| Montichiari | 3 |
| Montirone | 3 |
| Pisogne | 1 |
| Pontoglio | 3 |
| Pralboino | 1 |
| Rovato | 3 |
| Villa Carcina | 1 |
| Fenegrò | 3 |
| Mariano Comense | 3 |
| Gombito | 3 |
| Piadena | 3 |
| Stagno Lombardo | 3 |
| Colico | 1 |
| Valgreghentino | 1 |
| Brembio | 1 |
| Cavenago D'Adda | 1 |

**Table 5.** *Cont.*

| City | Monitoring frequency |
|------|:--------------------:|
| Orio Litta | 3 |
| Monza | 3 |
| Busto Garolfo | 1 |
| Gorgonzola | 1 |
| Milano | 3 |
| Milano | 3 |
| Pero | 3 |
| Borgo Virgilio | 3 |
| Castiglione Delle Stiviere | 3 |
| Cavriana | 3 |
| Goito | 3 |
| Gonzaga | 3 |
| Mariana Mantovana | 3 |
| Marmirolo | 3 |
| Medole | 3 |
| Pieve Di Coriano/Borgo Mantovano | 3 |
| Quistello | 3 |
| Rivarolo Mantovano | 1 |
| Sabbioneta | 3 |
| Solferino | 3 |
| Viadana | 3 |
| Ferrera Erbognone | 3 |
| Albonese | 1 |
| Travacò Siccomario | 1 |
| Vidigulfo | 1 |
| Samolaco | 1 |
| Arcisate | 1 |
| Busto Arsizio | 3 |
| Gerenzano | 3 |
| Somma Lombardo | 3 |

**Table 6.** Perfluorinated analyses in surface and ground water samples.

|  | PFBA | PFPeA | PFBS | PFHxA | PHFpA | PFHxS | PFOA | PFNA | PFDA | PFOS | PFUnA | PFDoA | TOT |
|------|------|-------|------|-------|-------|-------|------|------|------|------|-------|-------|------|
| **Analyses** | 416 | 416 | 416 | 416 | 416 | 416 | 416 | 416 | 416 | 416 | 416 | 416 | 4992 |
| **Surface water** | 286 | 286 | 286 | 286 | 286 | 286 | 286 | 286 | 286 | 286 | 286 | 286 | 3432 |
| **Ground water** | 130 | 130 | 130 | 130 | 130 | 130 | 130 | 130 | 130 | 130 | 130 | 130 | 1560 |
| **N < LOQ** | 246 | 280 | 284 | 279 | 339 | 405 | 235 | 395 | 405 | 98 | 416 | 416 | 3798 |
| **N > LOQ** | 170 | 136 | 132 | 137 | 77 | 11 | 181 | 21 | 11 | 318 | 0 | 0 | 1194 |

*3.1. PFAS Assessment in Ground Water*

Regarding ground water, eight of the twelve compounds monitored were detected in the area in concentrations above the Limit of Quantification (see Table 7).

In 87% of cases, the PFAS concentrations in groundwater were below the quantification limit (LOQ). In the cases where the quantification limits have been exceeded, the values found were lower than the minimum values set by the Italian regulation.

The most frequent findings are attributed to the PFOS and, in smaller numbers, to the PFOA and PFBS. In detail, PFOS was detected in 51% of the samples (LOQ = 0.2 ng L$^{-1}$), although always at concentrations below the limit value of equal to 30 ng L$^{-1}$ (see Table 3 and Figure 2). Sampling points are divided into 5 classes, not coinciding with the normative values". Moreover, for 41% of the analyzed samples (53 samples), PFOS was detected at levels higher than 0.65 ng L$^{-1}$; however, in no case PFOS value has exceeded the legal limit for groundwater.

**Table 7.** Perfluorinated distribution in ground water.

|  | PFBA | PFPeA | PFBS | PFHxA | PHFpA | PFHxS | PFOA | PFNA | PFDeA | PFOS | PFUnA | PFDoA | TOT |
|---|---|---|---|---|---|---|---|---|---|---|---|---|---|
| **Determinations** | 130 | 130 | 130 | 130 | 130 | 130 | 130 | 130 | 130 | 130 | 130 | 130 | 1560 |
| **N < LOQ** | 105 | 112 | 107 | 111 | 118 | 130 | 88 | 126 | 130 | 64 | 130 | 130 | 1351 |
| **N >= LOQ** | 25 | 18 | 23 | 19 | 12 | 0 | 42 | 4 | 0 | 66 | 0 | 0 | 209 |

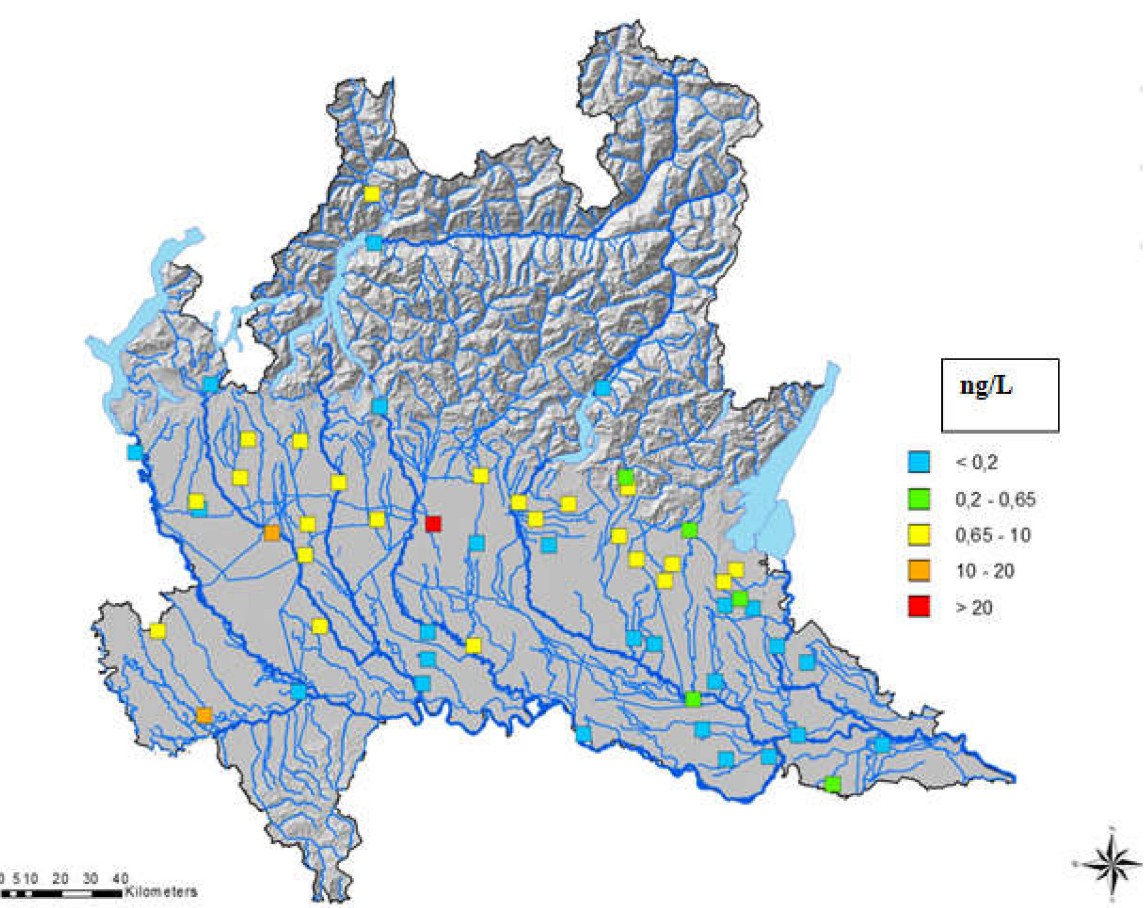

**Figure 2.** PFOS distribution in ground water samples in Lombardia Region.

*3.2. PFAS Assessment in Surface Water*

PFAS were detected in 44% of analyzed samples with PFOS and PFOA the most representative pollutants (71% and 44% detection, respectively). In most of cases, with the exception of PFOS, PFASs concentrations were below the limits of quantification and, in the cases where the limits of quantification were exceeded, the values found were lower than the legal limits (Legislative Decree No. 172/2015).

The percentages of findings for the other compounds present were: 44% for PFOA, 41% for PFBA and about 30% for PFBS, PFPeA and PFHxA.

Regarding perfluorooctansulfonic acid (PFOS), 228 exceedances of the standard envisaged by the legislation (equal to 0.65 ng L$^{-1}$) were found; in detail, 90% of the quantified samples showed the exceedance. On the other hand, it should be noted that all the values found for PFOS respect the maximum SQA-CMA concentration (36,000 ng L$^{-1}$).

For the PFOA and PFBS compounds, 2 and 3 exceedances were found, respectively. These numbers equal 1.5% of the PFOA findings and 3% of the PFBS findings.

In detail, (PFOA was found in Legnano (401 ng L$^{-1}$) and Rho (315 ng L$^{-1}$) while PFBS was found in Legnano (16000 ng L$^{-1}$), Rho (15000 ng L$^{-1}$) and Pero (3400 ng L$^{-1}$). However, these data not are

worrying, considering that for PFBS the EC50 tested by Daphnia magna in 48 h of exposure are 2183 and 477 ng L$^{-1}$ for PFBS and PFOA, respectively.

A similar distribution framework was obtained by following the assessments during the annual monitoring activities. In fact, each station was monitored several times during 2018, based on the frequencies as shown in Tables 1 and 2.

For PFBA, PFPeA and PFHxAno values exceeding the annual average concentration were detected.

The annual concentration average calculated for the PFBS has exceeded the SQA-MA value (3,000 ng L$^{-1}$) in 2 of the 55 monitored stations. Similarly, the average annual concentration calculated for the PFOA has exceeded the SQA-MA value (100 ng L$^{-1}$) in 1 of the 57 stations (see Table 8).

**Table 8.** Number of samples that are above the LOQ.

|  | PFBA | PFPeA | PFBS | PFHxA | PHFpA | PFHxS | PFOA | PFNA | PFDeA | PFOS | PFUnA | PFDoA |
|---|---|---|---|---|---|---|---|---|---|---|---|---|
| **Determinations** | 57 | 57 | 57 | 57 | 40 | 54 | 57 | 53 | 55 | 57 | 57 | 57 |
| **n < LOQ** | 9 | 27 | 35 | 30 | 40 | 54 | 27 | 53 | 55 | 3 | 57 | 57 |
| **n >= LOQ** | 48 | 30 | 22 | 27 | 0 | 0 | 30 | 0 | 0 | 54 | 0 | 0 |

A different picture concerns the PFOS compound must be considered: the average annual concentration calculated for perfluorooctane sulphonic acid has exceeded the SQA-MA limit value (0.65 ng L$^{-1}$) in 46 of the 57 stations monitored (81% of the cases) as shown in Figure 3.

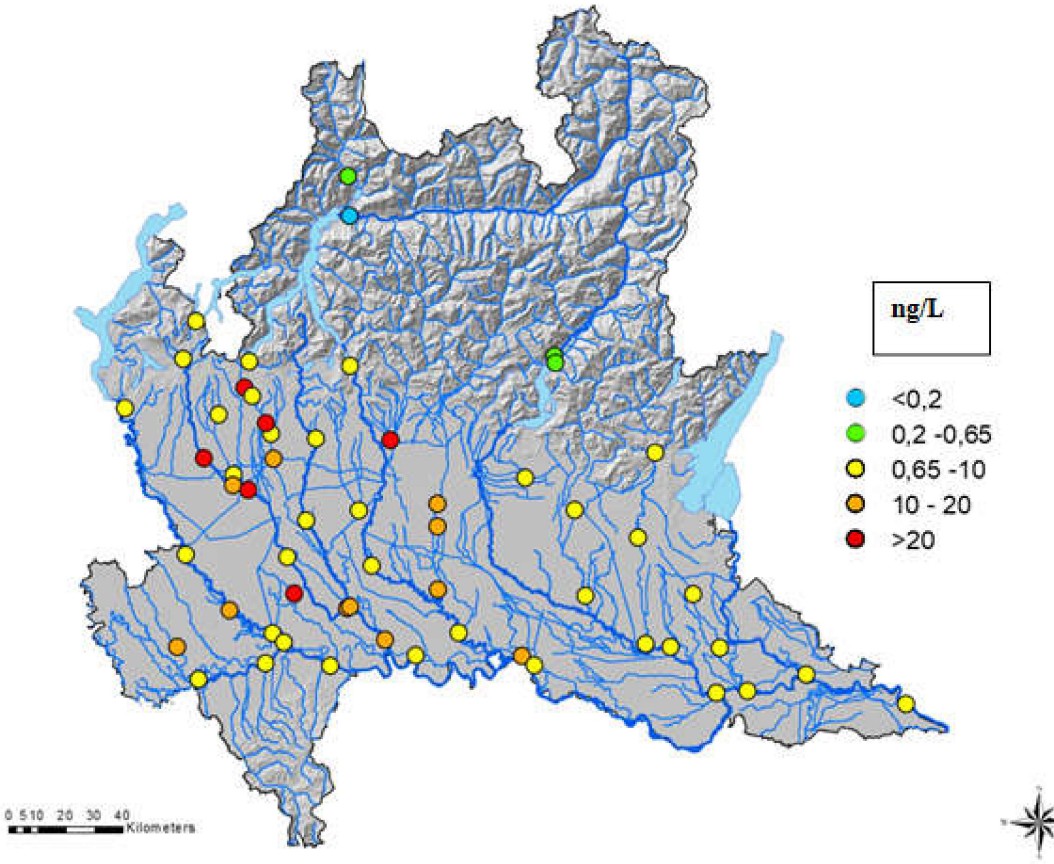

**Figure 3.** PFOS distribution in surface water samples in Lombardia Region. Sampling points are divided into five classes, not coinciding with the normative values.

The maximum concentration of PFOS was found in the sampling carried out in June in Legnano station on the Olona river (29.4 ng L$^{-1}$). Comparable values were found on the Olona stream (22 ng L$^{-1}$), Rho (17 ng L$^{-1}$) and Pero (21 ng L$^{-1}$) on the Olona river.

The distribution of concentrations for PFOS and PFOA is not very dissimilar between surface and underground waters (the main percentiles—calculated only on numerical values (>LOQ)—show quite similar concentrations (see Table 9).

Concerning PFOA, the maximum concentration values were found in the two stations of Legnano (401 ng L$^{-1}$) and Rho (315 ng L$^{-1}$) on the Olona river in the August 2018 sampling (Figure 4).

In Figure 3 the sampling points where PFOA were found at highest concentrations. Sampling points are divided into three classes, not coinciding with the normative values, with the only purpose of being able to have a picture of greater detail.

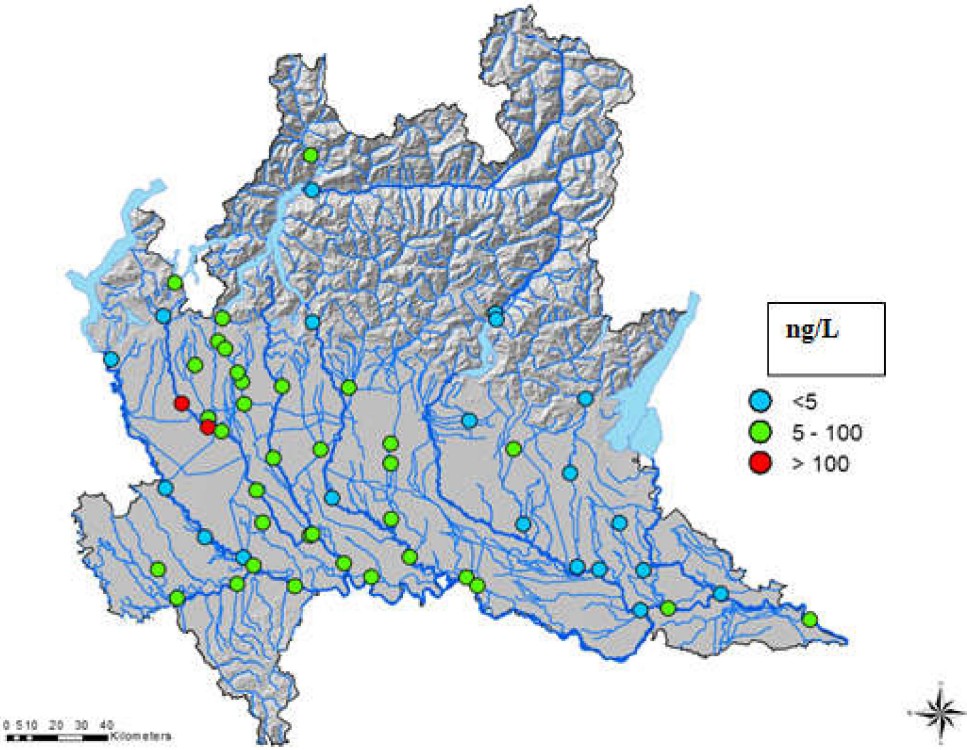

**Figure 4.** PFOA distribution in surface water samples in the Lombardia Region.

The maximum concentration values of PFBS were found in the three stations of Legnano, Rho and Pero on the Olona river in the August 2018 sampling.

**Table 9.** PFOS and PFOA distribution in surface and ground water samples.

| PFOS | Surface Water | Ground Water | PFOA | Surface Water | Ground Water |
|---|---|---|---|---|---|
| N analyses | 286 | 130 | N analyses | 286 | 130 |
| N. >= LOQ | 252 (88%) | 66 (51%) | N. >= LOQ | 139 (49%) | 42 (32%) |
| LOQ <= N. <= 0,65 ng/L | 24 (8%) | 13 (10%) | LOQ <= N. <= 100 ng/L | 137 (48%) | 42 (32%) |
| N. > 0,65 ng/L | 228 (80%) | 53 (41%) | N. > 100 ng/L | 2 (1%) | 0 |
| N. > 30 ng/L | 0 | 0 | N. > 500 ng/L | 0 | 0 |
| max (ng/L) | 29,4 | 23,2 | max (ng/L) | 401 | 67 |

Data analyses highlight that the distribution of PFCs in Lombardia region is similar if compared with other investigated areas that show a predominant contribution from PFOS rather than PFOA, indeed, PFOS concentrations are higher than those of PFOA in most of the sampling stations.

PFAS levels and distributions obtained, both for surface and ground waters, are in good agreement if compared to other studies carried out in Italy, Europe and extra Europe region by different researchers [16,17], but relatively lower than that found in groundwater in Changshu China in 2014 [18].

## 4. Conclusions

Perfluoroalkyl substances have only recently been introduced into Italian legislation concerning the quality of surface and groundwater water, with the inclusion of six compounds in the tables attached to Legislative Decree 172/2015 for surface water and four compounds in those attached to the DM 6 July 2016 for groundwater. The activity carried out and underway in 2018 is the first systematic monitoring at regional level carried out by ARPA Lombardia. From the first results, it is possible to draw a picture of the presence of these substances in the territory. For watercourses, the presence of perfluoroalkyl substances in the regional territory mainly concerns PFOS and PFOA and to a lesser extent PFBS and PFBA. The territory most affected by the presence appears to be that of the western plain.

For groundwater, on the other hand, the highest maximum levels of PFOS are found in the high plain, to which also the major findings of PFOA are to be added. Significant values of PFOA were also determined in the middle Pavia plain. The PFHxA and PFPeA compounds are concentrated in the high plain area of the Ticino-Adda Basin, while PFBA and PFBS are mainly present in the high Plain between Oglio and Mella and in the middle plain between Adda and Oglio.

Moreover, high-performance liquid chromatography–tandem mass spectrometry has proved to be a very good hyphenated technique able to detect low concentrations of pollutants in surface and groundwater samples.

**Author Contributions:** methodology, S.B. and M.B.; software, L.T. and V.M.; validation, S.B., M.B. and L.C. (Luisa Colzani); formal analysis, S.B., M.B.; investigation, L.T. and V.M.; resources, P.D. and L.C. (Laura Clerici); data curation, V.M.; writing—original draft preparation, S.B.; writing—review and editing, S.B. and D.D.; visualization, V.M. and L.T.; supervision, L.C. (Luisa Colzani), P.D. and L.C. (Laura Clerici); project administration, P.D.; funding acquisition, P.D. and L.C. (Laura Clerici). All authors have read and agreed to the published version of the manuscript.

**Funding:** The study was supported by Regione Lombardia financial source.

**Acknowledgments:** This work was performed thanks to a Regione Lombardia financial source.

**Conflicts of Interest:** The authors declare no conflict of interest.

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
