# Peer review of "Hyphenated High Performance Liquid Chromatography–Tandem Mass Spectrometry Techniques for the Determination of Perfluorinated Alkylated Substances in Lombardia Region in Italy, Profile Levels and Assessment: One Year of Monitoring Activities During 2018"

_separations, doi:10.3390/separations7010017_

Round 1

Reviewer 1 Report

As indicated, the paper requires serious and significant editing for English grammar and style. 

The information presented in the materials and methods section is inadequate in this reviewer's opinion. The entire manuscript is based on the reporting of the number of PFAS above/equal to, and below, the LOQ, yet the authors do not define the LOQ in this paper. The details on the MS parameters are also lacking. The authors have referenced other studies, but I would recommend including this information and moving the data on the number of samples, station IDs, etc.into the supplementary materials. Also, did the authors define and LOD? The number detected, but not quantified, might add to the context of the paper. Although the paper is presented as a methods paper, very little discussion of the method performance is presented. 

Table and figure captions are not sufficiently informative. No units are presented in the figure captions. 

There is really no interpretation done for the albeit large data set. Given the lack of information on the methods used and the lack of interpretation of the results, the paper is really more suited for a data repository entry as compared to a manuscript. If the authors added more detail in either (preferably both) of the areas described above, and reduced the amount of granular detail (Tables 2A and 2B, at minimum), this could be an informative study. 

Author Response

Dear Editor,

the manuscript entitled “Hyphenated high performance liquid chromatography-tandem mass spectrometry techniques for the determination of Perfluorinated Alkylated Substances in Lombardia Region in Italy, profile levels and assessment: one year of monitoring activities during 2018submitted by Salvatore Barreca, Maddalena Busetto, Luisa Colzani , Laura Clerici, Valeria Marchesi, Laura Tremolada and Pierluisa Dellavedova has been fully revised taking into consideration all the Reviewer’s comments.

We are thankful to the Reviewers who pointed out the weak points that needed to be improved.

Detailed answers to each of the Reviewer’s comments and consequential changes in the manuscript are listed below.

Review 1

Reviewer comment 1

As indicated, the paper requires serious and significant editing for English grammar and style.

Answer 1

As suggested, the paper was completed revised for English grammar and style

Reviewer comment 2

The information presented in the materials and methods section is inadequate in this reviewer's opinion. The entire manuscript is based on the reporting of the number of PFAS above/equal to, and below, the LOQ, yet the authors do not define the LOQ in this paper. The details on the MS parameters are also lacking. The authors have referenced other studies, but I would recommend including this information and moving the data on the number of samples, station IDs, etc.into the supplementary materials. Also, did the authors define and LOD? The number detected, but not quantified, might add to the context of the paper. Although the paper is presented as a methods paper, very little discussion of the method performance is presented.

Answer 2

The section material and methods has been implemented and the LOQ values for each analyte was added in Table 2.

Moreover, several MS parameters were added by table and data sampling station were added.

Regarding to Limit of Detection, by considering that PFAS data provided from Environmental Protection Agency of Lombardia Region in Italy are refer to European and Italian Decision, LOD not represent an important set required from Directive. For this reason, LODs have not been reported. On the other hand, discussion on the method performance was employed.

Reviewer comment 3

Table and figure captions are not sufficiently informative. No units are presented in the figure captions. 

Answer 3

Tables were improved and figure captions were revised by units.

Reviewer comment 4

There is really no interpretation done for the albeit large data set. Given the lack of information on the methods used and the lack of interpretation of the results, the paper is really more suited for a data repository entry as compared to a manuscript. If the authors added more detail in either (preferably both) of the areas described above, and reduced the amount of granular detail (Tables 2A and 2B, at minimum), this could be an informative study. 

Answer 4

Several interpretations were added in discussion section and some detail in new Table 3A and Tabe3B were removed.

Reviewer 2 Report

The investigation presented is original and inteseting. However, there are two aspect that new atention for international iimpact, and for its publication: 

1) The condicions of the method employed is not described, the authors refer to a reference however the reference does not include the information. I think the authors changed the reference list and missed this reference. 

2) The article does not have discussion and does not indicate the importance of the finding. Therefore i recomend to discusse the results and include some discusion in the finding in the abstract. What are the possible impact of the presece of this substances in the water. 

Author Response

Dear Editor,

the manuscript entitled “Hyphenated high performance liquid chromatography-tandem mass spectrometry techniques for the determination of Perfluorinated Alkylated Substances in Lombardia Region in Italy, profile levels and assessment: one year of monitoring activities during 2018submitted by Salvatore Barreca, Maddalena Busetto, Luisa Colzani , Laura Clerici, Valeria Marchesi, Laura Tremolada and Pierluisa Dellavedova has been fully revised taking into consideration all the Reviewer’s comments.

We are thankful to the Reviewers who pointed out the weak points that needed to be improved.

Detailed answers to each of the Reviewer’s comments and consequential changes in the manuscript are listed below.

Review 2

The investigation presented is original and inteseting. However, there are two aspect that new atention for international impact, and for its publication: 

Reviewer comment 1
1) The condicions of the method employed is not described, the authors refer to a reference however the reference does not include the information. I think the authors changed the reference list and missed this reference.

Answer 1

Details on method development were added in the paper.

Reviewer comment 2

2) The article does not have discussion and does not indicate the importance of the finding. Therefore i recomend to discusse the results and include some discusion in the finding in the abstract. What are the possible impact of the presece of this substances in the water.

Answer 1

Discussion section was implemented and abstract was revised.

Round 2

Reviewer 1 Report

The manuscript has been revised, however, the english quality is still poor. In many cases, this makes it difficult to understand what the authors did or have concluded. 

The authors have made efforts to address this reviewer's comments, but these efforts still fall short. The authors have provided MS parameters, but these parameters have no units. The authors still have not defined their calculation of LOQ (this value is critical to understanding the data in the paper and so the reader should not have to go to a reference to retrieve it. 

The paper still reads as a little sloppy-the lack of units for the MS parameters described above, in Table 4, limit values are listed as either "-" or "0"-what is the difference? 

There is more interpretation, but it is still lacking. 

Author Response

Dear Editor,

we thank the reviewer for assessing our work and for his/her valuable comments, which have allowed us to further improve our manuscript. Our response to each comment is as below.

Reviewer 1

The manuscript has been revised, however, the English quality is still poor. In many cases, this makes it difficult to understand what the authors did or have concluded.

Authors

English has been careffully revised.

Reviewer 1

The authors have made efforts to address this reviewer's comments, but these efforts still fall short. The authors have provided MS parameters, but these parameters have no units.

Authors

MS parametes units are been added

Reviewer 1

The authors still have not defined their calculation of LOQ (this value is critical to understanding the data in the paper and so the reader should not have to go to a reference to retrieve it.

Authors

A LOQ method calculation has been added in pag 5 line 143

“The quantification limits (LOQs) were estimated by 10-time standard deviation calculated at first level of calibration curve obtained in surface water sample”

Reviewer 1

The paper still reads as a little sloppy-the lack of units for the MS parameters described above, in Table 4, limit values ​​are listed as either "-" or "0" -what is the difference?

Authors

In all tables - was changed in 0.

Reviewer 1

There is more interpretation, but it is still lacking.

Authors

Several aspects were clarify in the document.

Reviewer 2 Report

The manuscript has been revised by the editor howeve I have the feelling tha the revision conducted by the authors has not been done in details, this can be clearly seen with the following points:
1) The method employed for the analysis has not been validated, validation conditions have not been included. It is not clear how accuracy, reproducibility and repeatibility has been evaluated.
The authors refers to reference 18 (which is not included in the list of referencenad reference 15 that it is an ISO).

2) Disscusion has not been included, the authors have only added to reference. Results are not discussed there are presented but not disccussed. Again information about the finding are not inlcuded.

Additionally the reviewer included question and comments in a doccuments and the authors have not replay to these coments.

I recoment a major carful revision, to be able to accept the manuscript for publication.

Author Response

Dear Editor,

we thank the reviewer for assessing our work and for his/her valuable comments, which have allowed us to further improve our manuscript. Our responses to each comment are reported below.

Reviewer 2

The manuscript has been revised by the editor howeve I have the feelling tha the revision conducted by the authors has not been done in details, this can be clearly seen with the following points:

The method employed for the analysis has not been validated, validation conditions have not been included. It is not clear how accuracy, reproducibility and repeatibility has been evaluated.

Authors

Revision has been carefully conducted

Reviewer 2

The authors refers to reference 18 (which is not included in the list of referencenad reference 15 that it is an ISO).

Authors

Reference was adjusted

Reviewer 2

2) Disscusion has not been included, the authors have only added to reference. Results are not discussed there are presented but not disccussed. Again information about the finding are not inlcuded.

Additionally the reviewer included question and comments in a doccuments and the authors have not replay to these coments.

I recoment a major carful revision, to be able to accept the manuscript for publication.

Authors

Several parts were added and careful revision was made as following

Reviewer 2 Page 1 line 26 hold

The order of the two sentences were changed as following

Authors Page 2 line 25 new 

Among the results, PFOS, PFOA, PFBA, PFBS, PFPeA and PFHxA were identified as the most abundant analytes detected. PFASs concentrations, in most cases, were below the limits of quantification and, in the cases where the limits of quantification have been exceeded, the values found were lower than Italy directive. PFOS is an exception and in fact this compound was detected in 76% of analyzed samples (surface and ground waters).

Reviewer 2 Page 2 line 67 hold

Sentence was adjusted as suggested

Authors Page 2 line 74 new 

In detail, the method consists in a concentration by online solid phase extraction coupled with ultra-high-performance liquid chromatography-tandem mass spectrometry, in order to quantify PFASs at very low limit of quantifications [14].

Reviewer 2 Page 2 line 83  hold

Sentence was modify as suggested

Authors Page 3 line 94 new

High purity water was prepared using a Millipore Milli-Q purification system.

Reviewer 2 Page 2 line 86 Page 3 line 89 and page 3 line 85 hold

Authors

City were added

Reviewer 2 Page 3 line 106,

Authors Page 4 line 123

References was changed from 18 to 14

Reviewer 2  Page 3 Table 1B

Authors Table 1B was revised as Reviewer suggested

Reviewer 2 Page 4 line 112

Authors sentence was modified

 Samples were collected in a 50 mL Polypropylene tube (purchased from CPS Analitica for Chemistry, Milan, Italy).

Ten mL of samples were transferred to a 10 mL vial, then 0,1 mL Internal Standard (IS) solutions at 5000 ng/L were added in order to obtain a final IS concentration of 50 ng/L; they were manually mixed and stored in a refrigerated autosampler system at 4 °C. 5 mL of sample was injected by using a SPE online system as discussed in previous paper [14]. The samples were injected directly without filtration step.

Reviewer 2 Page 4 line 118

Authors several information were added

The analyses were performed by using a method developed and previously reported in scientific literature [14].

Briefly, method performances were evaluated by estimating quantification limits (LOQs) and linearity.

Method was validated in term of quantification limits, accuracy and precision as directive required [15]. Analyses were conducted by using real surface water samples.

The quantification limits (LOQs) were estimated by 10-time standard deviation calculated at first level of calibration curve obtained in surface water sample. 

Table 2 summarizes the results obtained for these performance parameters. Accuracy and repeatability were calculated on data set of six analyses conducted on water spiked samples.

Linearity ranges were obtained from 0.2-250 to 5-250 ng L-1 depending on the different analyte considered.

PFBA shows a linear range from 5 to 250 ng L-1. Due to its short alkyl chain, chemical physical properties of PFBA are very different compared to other PFASs compounds. In this regard, PFBA is adsorbed less than other PFASs in SPE online and in this context the first level corresponding to 5 ng L-1. In any cases, by comparing this method with other direct injection or SPE methods, a good detection limit for PFBA was achieved. For the most PFASs, good correlation can be obtained in calibration curve range from 1 to 250 ng L-1.

Reviewer 2 Page 5 line 134.

Authors A Figure 1 with sample locations was added and mistakes were adjusted

Reviewer 2 Page 9 line 160

Sentence was changed in

Authors page 11 line 213

 In most of cases, with the exception of PFOS, PFASs concentrations were below the limits of quantification and, in the cases where the limits of quantification have been exceeded, the values found were lower than the legal limits (Legislative Decree No. 172/2015).

Reviewer 2 Page 9 line 172

Authors pag 11 line 227  Sentence was modify as reported

In detail, (PFOA was found in Legnano (401 ng/L) and Rho (315 ng/L) while PFBS was found in Legnano (16000 ng/L), Rho (15000 ng/L) and Pero (3400 ng/L). However, these data not are worrying considering that for PFBS the  EC50 tested by  Daphnia magna in 48 h of exposure are 2183 and 477  ng/L for PFBS and PFOA respectively

Reviewer 2 Table 5 title

Authors 

The title of table 5 was adjusted and Figure 3 was mentioned in the text.
